# Radiation-Activated Pre-Differentiated Retinal Tissue Monitored by Acoustic Wave Biosensor [note 1]

**DOI:** 10.3390/s20092628

**Published:** 2020-05-05

**Authors:** Alin Cheran, Michael Thompson

**Affiliations:** Department of Chemistry, University of Toronto, 80 St. George Street, Toronto, ON M5S 3H6, Canada; alin.cheran@sinai.org

**Keywords:** thickness-shear mode acoustic wave biosensor, retinal tissue, radiation activation

## Abstract

A thickness-shear mode acoustic wave biosensor operated within a flow-through system was used to examine the response of mouse retinal tissue to radiation. Control experiments conducted with respect to exposure of the bare gold electrodes of the device under various conditions of light intensity and bathing solution yielded reversible changes in resonant frequency (Fs) and motional resistance (Rm). The magnitude of transient changes was proportional to light intensity, but independent of solution type. These alterations in acoustic parameters were ascribed to acoustic coupling phenomena at the electrode-to-liquid interface. Pre-differentiated retina from mouse samples deposited on the thickness shear mode (TSM) electrode exposed to a high light intensity condition also exhibited reversible changes in both Fs and Rm, compared to control experiments involving a coating used to attach the tissue to the electrode. In this case, the radiation-instigated reversible responses for both acoustic parameters exhibited a reduction in magnitude. The changes are ascribed to the alteration in viscoelasticity of the retinal matrix on the TSM electrode surface. The precise biophysical mechanism responsible for the changes in Fs and Rm remains a challenge, given the complex make up of retinal tissue.

## 1. Introduction

The study of biological entities such as proteins, nucleic acids, cells, and tissue at the solid–liquid interface by acoustic wave (AW) biosensors has attracted considerable attention in recent years [1]. Such fundamental studies of a population of cells, in vitro, improves our basic knowledge and can provide a pillar for understanding the function of much more complex systems, such as biological organs. A key issue which is common to many studies involving AW devices is the necessity to attach and proliferate cells on inorganic surfaces, such as gold and silicon [2]. This process often requires the use of coatings or alternative methods of surface modification in order to allow cell adhesion. In an attempt to mimic the extra-cellular matrix, cocktails of moieties such as laminin, elastin, and fibronectin are often used as coatings. A further critical factor is the necessity to operate the sensor under conditions which are capable of the maintenance of cell viability and function. 

The thickness shear mode (TSM) device offers the ability to study cells at acoustic wavelengths, non-invasively, under label-free conditions and in real-time. The sensor was employed over the years in the study of a variety of cells, including endothelial cells, kidney cells, cancer cells epithelial cells fibroblasts, osteoblasts, and smooth muscle cells [3,4,5,6,7,8,9,10,11,12]. Research on cells was also conducted with other devices, such as Love and surface wave sensors, and on detailed examination of cell behavior, beautifully reviewed in Reference [12], where a common feature is the issue of surface adherence of cells on a number of substrates, such as titanium and hydroxyapatite. Included with the latter are often investigations of viscoelastic properties of cells, as influenced by growth factors and small molecules such as drugs. 

In our own research, we concentrated on acoustic wave study of hypothalamic neurons prepared from mouse embryo tissue at a confluence in the range of 80–100% [13,14,15,16]. Experiments were conducted with both KCl and NaCl, in terms of interaction with neurons in a flow-injection configuration, with measurement of changes in resonant frequency and motional resistance. The various responses to these electrolytes were interpreted in terms of changes in cellular structure associated with the depolarization process. With regard to adhesion and deposition of neurons, in the absence of serum proteins, initial cell adhesion was followed by subsequent cell death and removal from the sensor surface. Further, the presence of the peptide, GRGDS (Gly-Arg-Gly-Asp-Ser peptide) was observed to significantly reduce cell-surface specific interactions, compared to the control of SDGRG (Ser-Asp-Gly-Arg-Gly peptide). Finally, we investigated the synchronization of the circadian rhythm generator and the effects of glucagon on hypothalamic neurons. For full synchronization, the addition of the serum bolus triggered both increases in resonant frequency and motional resistance, which decayed after 30 min. The duration of this decay closely matched the time required for full synchronization.

The object of this work was to perform a proof of concept experiment, where we could show that pre-differentiated tissue, which is far more complex in terms of biological architecture than previous studies of cell populations, could be interrogated with the use of the TSM. Unlike the situation with studies of cell populations by AW devices, examination of mature tissue has not featured prominently in the literature. In this case, we chose to explore the behavior of retinal tissue excised from sacrificed white mice which was deposited on the electrode surface of a TSM device. In terms of the future, such experiments could lead to a correlation of tissue biophysical processes with, for example, treatment with drugs. This approach was employed by our group in an examination of the response of hypothalamic neurons to neurotrophic factors [17]. Using careful control of temperature and acoustic cell flow conditions, the tissue samples were exposed to various intensities of white light. Changes in both resonant frequency and motional resistance were observed on tissue exposure to radiation.

## 2. Experimental

### 2.1. Attachment of Retinal Tissue to TSM Device

Nine MHz AT-cut quartz crystals (13.4 mm diameter) with symmetrical gold electrodes (4.9 mm diameter) obtained from LapTech. Corp. (Bowmanville, ON, Canada) were incorporated into a flow-cell system equipped with a white light source. The sensors were sonicated and washed in 1% w/w SDS, methanol, and acetone for five minutes in each solvent. Before use, crystals were treated with sterile water and serum-free Dulbecco’s Modified Eagle Medium (DMEM). Clean sensors were coated with a cell attachment matrix consisting of an entactin, collagen, and IV-laminin mixture (ECL) from Upstate Biotechnology, Inc. (Waltham, MA, USA) at 5 mg cm^−2^ for 2 h. Retinal tissue obtained from sacrificed white mice was applied to the middle of the sensor, with photoreceptors facing up in DMEM supplemented with 5% fetal bovine serum (Hyclone, GE Healthcare Life Sciences (Pittsburgh, PA, USA)). The preparation was encased in a sample holder and allowed to equilibrate for 30 min with no flow-through at 37 °C.

### 2.2. Procedure

The assembled instrument with the prepared sensor was allowed to equilibrate to 37 °C, with establishment of a baseline with a flow-through of 5% FBS DMEM at 50 uL/min for several hours. The retinal tissue was exposed to a high intensity of light, with the behavior of the tissue being monitored by the TSM sensor, with respect to frequency as well as motional resistance every 8 s. Control studies were performed using bare sensors and devices coated with the cell-attachment matrix noted above. Both conditions were exposed to a flow-through of distilled water, phosphate buffered saline (PBS), serum-free DMEM, and 5% FBS DMEM at a rate of 50ul/min. In addition, each flow-through solution was exposed to three intensities of light (low, medium, and high) with off-phases in between. Each phase was typically 5–10 min in duration. The flow-through rate was unchanged for the duration of the experiments, only light intensities were altered without disturbing the flow-through chamber. The bottom chamber was exposed to the outside environment (air) being kept at 37 °C and constant humidity. Three experiments were conducted with different retinal preparations.

## 3. Results and Discussion

### 3.1. Acoustic Wave Propagation into Substrate-Attached Tissue

Prior to presenting the results of this work, it is useful to concisely discuss the interaction of the acoustic propagated from the TSM device into the retinal tissue layered onto the electrode surface. The thickness of the retinal layer including the extracellular matrix (ECM) component is anticipated to be of the order of 100–500 µm [18]. When operated in liquids, the acoustic wave is dissipated into the liquid, with inertial storage changes being related to the change in frequency (Fs). Dissipation is measured as a change motional resistance (Rm), where, under liquid loading, the value is governed by the viscosity and density of the fluid. In liquids, the penetration depth of the wave originating from a 9 MHz device is of the order 0.2 µm. With cells including the ECM, the penetration depth of the wave is around 0.5 µm [19], which implies that a component of the cytoplasm will interact with the acoustic wave. For tissue, such as described in the present study, the Fs and Rm values will be controlled by the acoustic impedance of the composite of the various layers of the far thicker biological material, resembling the characteristics of a film of hydrogel. The long dimension of rod cells is 100 µm. Finally, the device in this case was not expected to function as a so-called “mass” sensor as described by Sauerbrey [20]; any alterations in Fs and Rm values were much more likely to be governed by viscoelastic changes and/or coupling of the biological sample to the electrode surface. 

### 3.2. Control Experiments–TSM Electrode Exposure to Radiation

The control experiments conducted in this work involved the bare TSM gold electrode being exposed to light of three different intensities in on- and off- modes. A typical plot for Fs and Rm under the high light condition is shown in Figure 1. Interestingly, the sensor yields reversible changes in both Fs and Rm, with increases in the former parameter and decreases in the latter both being observed. The instigated changes in both Fs and Rm for various bathing solutions were obviously dependent on the three intensities of light employed in the experiments. However, it was clear that solution type was of no influence on the acoustic response (Figure 2). Possible explanations for these changes are alterations in temperature of the bathing flowing medium, removal and/or addition of “mass”, or a manifestation of an acoustic coupling phenomena. The first two of these explanations can be readily discounted, since a reversible change in temperature of the flowing medium in the time frame observed and repeated gain/loss of mass are obviously highly unlikely. Furthermore, the temperature of the system was very carefully controlled during all experiments. 

Radiation-instigated changes in acoustic parameters were observed previously with regard to photooxidation of dithiol-containing monolayers on gold (TSM electrode) [21]. In these experiments, the monolayers on the TSM Au electrodes were exposed to varying levels of oxygen, followed by measurement of acoustic parameters on exposure to UV-radiation. Surface analysis of the monolayers by SIMS showed that the thiol groups were oxidized. The changes in Fs and Rm were ascribed to reversible coupling of the monolayer to the surrounding liquid associated with the photooxidation phenomena. It is highly likely that an analogous effect was observed in the present work, given that it is virtually impossible to ensure that TSM gold electrodes are contamination free, especially of S-containing moieties, unless high vacuum conditions are employed [22]. It has long been established that genuinely clean Au exhibits hydrophilic characteristics, rather than the usual surface of low polarity, which can occur despite ostensibly stringent cleaning protocols [23].

### 3.3. Acoustic Parameters for Exposure of Substrate-Attached Tissue to Radiation

A typical plot of Fs and Rm for retinal tissue ECL attached to the TSM electrode and exposed to a high light condition is depicted in Figure 3. For access to retinal tissue reasons, only the high intensity was employed in this experiment. DMEM was the sole solution used to bathe the retinal tissue. Using H_2_O would lyse cells, and the use of media without sufficient trophic factors such as PBS, although sufficiently buffered, may interfere with cellular metabolic processes during the experiment. This is the primary reason why only a solution rich in vitamins, amino acids, and glucose (DMEM) was used for retinal cell containing experiments.

It is noteworthy that in this case there is a reversible decrease in both Fs and Rm, unlike the situation for experiments conducted on the bare electrodes. A comparison of the changes for ECL-coated electrodes and those with tissue in place is shown in Figure 4. Clearly, the reversible alterations in the parameters are statistically significant, implying that the reductions in Fs in Rm are associated with alterations in inertial storage and dissipation, respectively. Changes in the same direction for the two parameters for TSM studies of an RNA- peptide interaction were interpreted in terms of both viscoelastic and acoustic coupling via slip phenomena [24]. The decrease in Fs implies an increase in viscosity; whereas, for Rm, the indication may be an increase in film stiffness. Additionally, conformational shifts in biological macromolecules imposed on TSM devices are known to cause perturbation of both Fs and Rm [25].

Ascribing a mechanism to the alterations in acoustic parameters in terms of the expected effect of radiation on surface-attached retinal tissue at this point in this research, represents something of an intractable problem. Concisely, mammalian retinal tissue consists of ten distinct layers, including the photoreceptor cells, the rods, and cones [18]. Transduction in these cells associated with exposure to radiation involves a cis-to-trans isomerization of the chromophore and retinal, which is a component of the light-sensitive protein, rhodopsin. A cascade of subsequent biochemical processes leads to a hyperpolarization of the cell membrane. The photoreceptor cells synapse with other neural components, such as bipolar and horizontal cells, which in turn connect to ganglion cells. An action pulse at the latter is then conveyed by nerve conduction to the brain. Some of these cells also display depolarization in addition to hyperpolarization. Relevant to the present study, we saw changes in Fs and Rm when mouse embryo neurons imposed on a TSM device were exposed to electrolytes [13]. In this case, such responses in acoustic parameters were ascribed to depolarization-instigated cellular structural changes, possibly involving alteration of viscoelasticity, or coupling of the neuronal layer to the ECL.

Accordingly, in the present study, given the reversibility in acoustic parameters in tandem with radiation on-and-off exposure, the device must be responding to the complex cascade of cellular events involved in the phototransduction process, which are clearly transitory. In this work, the photoreceptor rod cells, with an expected length of 100 µm, directly face the flowing solution. However, the aforementioned biochemical cascade is highly likely to proceed towards the electrode surface, thus influencing the acoustic properties of the complete retinal layer.

## 4. Concluding Remarks

The results of this work present two interdependent issues in terms of the use of acoustic wave study on samples of tissue held at an interface. First, as specified above, the interpretation of the alteration in acoustic parameters caused by the exposure of this particular sample to light remains a challenge. With respect to the monolayers of cells of various types, analogous research is most often concerned with the straightforward characterization of adhesion behavior on surfaces. In addition, there are many discussions of viscoelastic perturbations associated with interaction with stimuli, such as drugs, as indicated above for reference [17]. However, even for the “simpler” situation of research on cells, a precise correlation of changes in biochemical properties with acoustic parameters remains somewhat elusive. In the present work, the matrix is significantly more complex, where many bio-physicochemical processes are operating in a cascade involving a number of different cells. 

One potential application for acoustic wave technology, as for a number of biophysical techniques, is the phenomenological correlation of measured parameters obtained from healthy tissue extracted from animals with tissue that is malfunctioning. With specific regard to the retina, a number of such conditions are known, such as retinal detachment, diabetic retinopathy, and dry macular degeneration [26]. The first of these involves the separation of retinal cells from the layer of blood vessels that provides oxygen and nourishment; the second, damage to the blood vessels in the eye; and the last condition, thinning of the macula. All of these effects can result in a severe loss of vision, and research of this type would necessarily involve healthy and damaged tissue excised from animals. However, as mentioned above, a definitive characterization of the biology of these conditions though acoustic wave biosensor response would be highly dependent on a more solid interpretation of the responses of acoustic parameters to stimuli imposed on tissue. 

## Figures and Tables

**Figure 1 sensors-20-02628-f001:**
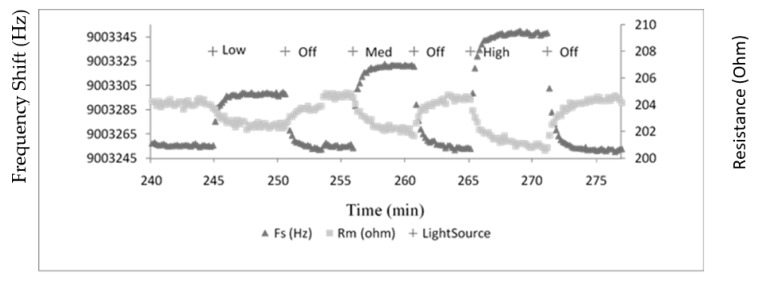
Typical plot of resonant frequency (Fs) and motional resistance (Rm) for exposure of bare Au electrodes of thickness shear mode (TSM) to various intensities of light. The system was allowed to equilibrate for several hours before light exposure.

**Figure 2 sensors-20-02628-f002:**
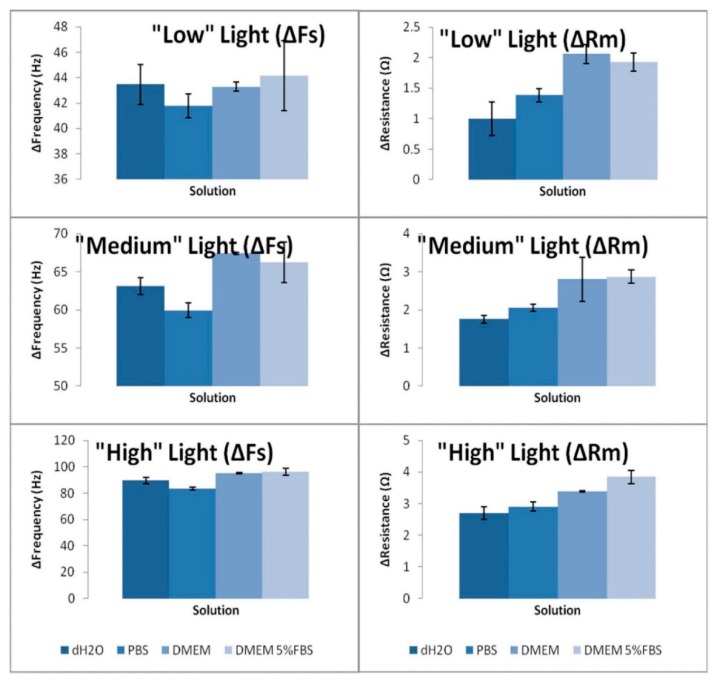
Changes in Fs and Rm for exposure of bare TSM electrodes to various solutions and light intensity conditions.

**Figure 3 sensors-20-02628-f003:**
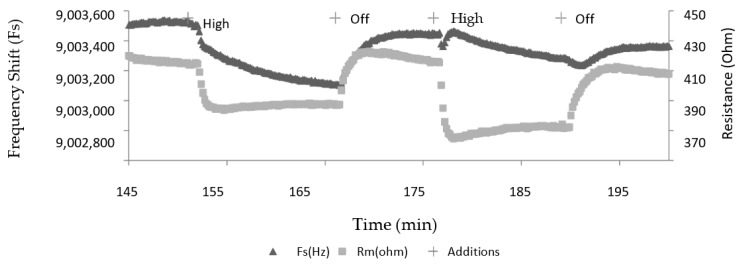
Changes in Fs and Rm after exposure of retinal tissue deposited on a TSM Au electrode under high light intensity condition.

**Figure 4 sensors-20-02628-f004:**
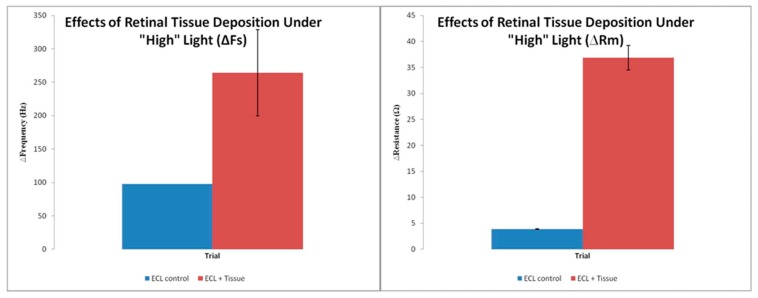
Changes in Fs and Rm for both ECL control experiments and retinal tissue deposited on a TSM Au electrode. High light intensity condition.

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
