# Peer review of "Radiation-Activated Pre-Differentiated Retinal Tissue Monitored by Acoustic Wave Biosensorâ€"

_sensors, 2020, doi:10.3390/s20092628_

Round 1
Reviewer 1 Report
The paper sounds consistent and could be published in the present form. My only concern is the conclusion.
The paper describes a sensor that detects some changes in the environment by the impact of light. The physical processes to create such responses in the acoustic resonator are more or less known, their origin from processes in the environment, however, remains in the dark. That is unsatisfying if the sensor should be used for “increasing our understanding of the basic mechanisms” (line 194/195) in tissues.
On the other side, for “the detection of malfunctioning tissue that can occur in disease states” (line 196/197) it is not clear, how such a sensor could be brought into contact to the tissue for the ‘animal-based investigations’ (line 199).
Reviewer 2 Report
Please check the attached file
